# Peroxyacetic Acid Pretreatment: A Potentially Promising Strategy towards Lignocellulose Biorefinery

**DOI:** 10.3390/molecules27196359

**Published:** 2022-09-26

**Authors:** Mingyang Hu, Junyou Chen, Yanyan Yu, Yun Liu

**Affiliations:** College of Life Science and Technology, Beijing University of Chemical Technology, Beijing 100029, China

**Keywords:** lignocellulose, peroxyacetic acid (PAA) pretreatment, mass balance, economic evaluation, biorefinery

## Abstract

The stubborn and complex structure of lignocellulose hinders the valorization of each component of cellulose, hemicellulose, and lignin in the biorefinery industries. Therefore, efficient pretreatment is an essential and prerequisite step for lignocellulose biorefinery. Recently, a considerable number of studies have focused on peroxyacetic acid (PAA) pretreatment in lignocellulose fractionation and some breakthroughs have been achieved in recent decades. In this article, we aim to highlight the challenges of PAA pretreatment and propose a roadmap towards lignocellulose fractionation by PAA for future research. As a novel promising pretreatment method towards lignocellulosic fractionation, PAA is a strong oxidizing agent that can selectively remove lignin and hemicellulose from lignocellulose, retaining intact cellulose for downstream upgrading. PAA in lignocellulose pretreatment can be divided into commercial PAA, chemical activation PAA, and enzymatic in-situ generation of PAA. Each PAA for lignocellulose fractionation shows its own advantages and disadvantages. To meet the theme of green chemistry, enzymatic in-situ generation of PAA has aroused a great deal of enthusiasm in lignocellulose fractionation. Furthermore, mass balance and techno-economic analyses are discussed in order to evaluate the feasibility of PAA pretreatment in lignocellulose fractionation. Ultimately, some perspectives and opportunities are proposed to address the existing limitations in PAA pretreatment towards biomass biorefinery valorization. In summary, from the views of green chemistry, enzymatic in-situ generation of PAA will become a cutting-edge topic research in the lignocellulose fractionation in future.

## 1. Introduction

Due to serious environmental issues and global climate change, researchers all over the world are trying their best to convert the fossil fuel-based society into a bio-economical society, advancing the goal of reaching peak carbon and realizing carbon neutrality [1,2]. Although fossil fuels play a critical role in social industrialization, these non-renewable and unsustainable fuels have negative effects on the environment and humans [3,4]. Lignocellulose, such as forest residues (branches, leaves, etc.), agricultural residues (wheat straw, rice straw, etc.), energy crops (willow, poplar, etc.), and cellulosic waste (e.g., municipal solid waste and food waste) are abundant and cost-effective renewable resources with an annual production of 15–17 × 10^10^ Mt [5,6]. Lignocellulose can be upgraded into biofuels, biochemicals, and biomaterials [7,8]. Therefore, lignocellulose biorefinery is expected to replace the traditional petroleum refining, and this will mitigate energy crisis and environmental pollution [9]. The United Nations Conference on Environment and Development (UNCED) predicts that the utilization of biomass resources may reach half of the world’s total resource use by 2050 [10].

However, pretreatment processes are required to destroy the stubborn structure of lignin, resulting in the improvement of the accessibility of cellulase to cellulose for the downstream utilization [11]. At present, four major methods of lignocellulose pretreatment are described in the literature [12]. Each method has its own advantages and disadvantages. For instance, physical pretreatment, such as milling and grinding, can improve the surface area and porosity of lignocellulose, but the high energy consumption of this pretreatment increases the operational costs and limits its practical applications [13]. Chemical pretreatment of dilute acids, bases, organic solvents, ionic liquids, and low eutectic solvents can remove lignin and hemicellulose to improve the enzymatic accessibility of cellulose, and can also reduce the degree of polymerization (DP) and crystallinity (Crl) of cellulose [14]. However, a critical issue in chemical pretreatment is that chemical reagents are expensive and prone to corrode equipment. Physico-chemical pretreatment is a combination of physical and chemical pretreatment; this method can dissolve lignin and hemicellulose to facilitate the utilization of cellulose [15]. Typical physicochemical pretreatment includes steam explosion, liquid hot water, ammonia fiber explosion, ammonia cycle permeation, electrocatalysis, CO_2_ explosion, and SO_2_ explosion [16]. The drawbacks of physicochemical pretreatment are that it requires high temperatures and high-pressure reaction conditions. Biological pretreatment uses microbial communities such as fungi or bacteria to damage the lignocellulosic structure. It is a novel pretreatment method with low energy consumption and low environmental impact [17]. However, an unsatisfactory aspect is that the low efficiency of biodegradation pretreatment limits its large-scale industrial applications [18].

Peroxyacetic acid (PAA), an organic peroxy acid, has been extensively regarded as a disinfectant, strong oxidizer, preservative, bactericide, and polymerization catalyst [19]. In recent years, PAA has been employed as a strong oxidant to oxidize the hydroxyl group in the lignin side chain to the carbonyl group, and it will cleave the β-aryl bond of lignin to reduce the molecular weight and introduce hydrophilic groups [20]. PAA will also oxidize the hydroxyl group in the lignin side chain to hydroquinone; it is subsequently oxidized to quinone, whose ring opening generates water-soluble hydroponic acid, maleic acid, and fumaric acid derivatives [20]. Through these reactions, lignin is depolymerized and the fragments will dissolve in water, leading to effective removal from lignocellulosic biomass [21]. In addition, the oxidized lignin shows low hydrophobicity and weakens the ability to bind to cellulase. Therefore, an increasing number of studies have been focusing on PAA pretreatment in lignocellulosic biorefinery.

In this article, we aim to highlight the challenges of PAA pretreatment and propose a roadmap towards future research into lignocellulose fractionation by PAA. We start with introducing of lignocellulose structure, and reviewing three kinds of PAA in lignocellulose pretreatment, including commercial PAA, chemical activation PAA, and enzymatic in-situ generation of PAA. Subsequently, the advantages and disadvantages of each PAA towards lignocellulose fractionation are extensively analyzed. To meet the theme of green chemistry, this article focuses on enzymatic in-situ generation of PAA and highlights its probable challenges in lignocellulose fractionation at this current stage. Furthermore, the mass balance and techno-economic feasibility of PAA pretreatment in lignocellulose fractionation are extensively discussed. At the end of the paper, critical perspectives and opportunities are proposed based on the existing limitations in PAA pretreatment towards biomass biorefinery valorization.

## 2. Lignocellulose Structure

Lignocellulose biomass is an abundant, diverse, and inexpensive renewable resource in nature. It has been universally converted into biofuels, biochemicals, and biomaterials [22]. As shown in Figure 1, lignocellulose is mainly composed of cellulose (40–45%), hemicellulose (20–40%), and lignin (10–25%), which are tightly bound together to form the skeletal framework of plant. The three-dimensional network structure shows that cellulose and hemicellulose are mainly connected by hydrogen bonds, and lignin and hemicellulose are also linked with chemical bonds, such as hydrogen bonds, ionic bonds, covalent bonds, and hydrophobic interactions [23].

### 2.1. Cellulose

Cellulose, the most abundant polymer on Earth, is a linear intercalation (alternating spatial arrangement of side chains) homopolymer. It consists mainly of β-(l-4) glycosidic bonds linked by alternating arrangements [24]. Due to its unique structure of ordered bundle arrangement and highly crystalline structure, cellulose is very stable in many conditions. Cellulose has good biocompatibility and active hydroxyl groups with an atomic O/C of 0.6–0.83 and H/C of 0.8–1.67 [25]. Cellulose can be valorized into fermented glucose [26], bioethanol [27], biomaterials [28,29], and catalyst carrier [30].

### 2.2. Hemicellulose

Hemicellulose has a heteropolymer with a relatively lower molecular weight compared to cellulose; it is composed mainly of pentoses (e.g., xylose and arabinose) and hexoses (e.g., mannose, glucose, and galactose) [31]. Hemicellulose is bound to various other cell wall components such as fibronectin, cell wall proteins, lignin, and phenolic compounds through covalent bonds, hydrogen bonds, and hydrophobic interactions [32]. Hemicellulose has been mainly used to produce fructose and xylitol. Apart from these products, hemicellulose can also be converted to biofuels [33], furfural [34], levulinic acid; and formic acid [35,36].

### 2.3. Lignin

Lignin is a polymer of heterogeneous phenyl propane units in plants and consists of three main monomers: guaiacol (G), eugenol (S), and p-hydroxyphenyl (H) [37]. These three monomers are chemically linked with the C-C bond (5-5, β-β, β-1, β-5) and aryl ethers (β-O-4, α-O-4) to yield three corresponding subunits: p-coumaryl alcohol (pCoumA), pineal alcohol (ConA), and mustard alcohol (SinA) [38]. Due to the heterogeneity and complex components, lignin shows strong stubborn and anti-barrier effects [39]. To date, lignin has mainly been used in reinforcing agents [40], binders [41], hydrogels [42], adsorbents [43], and catalysts [44]. Efficient valorization of lignin will be a hot topic of research in the near future.

## 3. Quick Overview of PAA

As mentioned above, in order to valorize each component of lignocellulose, pretreatment processes should be required to destroy its stubborn structure. To this end, a novel promising alternative, PAA pretreatment, is introduced in this work. First of all, we present a quick overview of PAA. As a strong oxidant, PAA is extensively used in wastewater disinfection due to its good disinfection performance and the low toxicity of its by-products [45]. Appendix A shows the chemical structure of PAA with a high oxidation potential (1.748 V) [46]. The O-O the bond dissociation energy of PAA (159 kJ·mol^−1^) is relatively weaker than that of hydrogen peroxide (213 kJ·mol^−1^) [47]. Three kinds of PAA are reported in the literature, including commercial PAA, chemical activation PAA, and enzymatic in-situ generation of PAA.

### 3.1. Commercial PAA

Commercial PAA products are greatly dependent on the ratio of PAA to hydrogen peroxide (H_2_O_2_). Table 1 provides detailed information on part commercial PAA in the literature. Commercial PAA is usually prepared by mixing H_2_O_2_ and acetic acid (or ethyl acetate), catalyzed with concentrated sulfuric acid. The desired concentration and yield of PAA are achieved by adjusting the concentration of H_2_O_2_ and the ratio of acetic acid. However, the chemical production of PAA is characterized by flammability, explosiveness, toxicity, high temperature, high pressure, and corrosiveness. From the point-of-view of safety and green chemistry, it is very dangerous to produce commercial PAA in the laboratory.

### 3.2. Chemically Activated PAA

To improve the oxidative ability of commercial PAA, some activators can be added to the PAA system. These activators include radiation, metal catalysts, and carbon-based materials [50,51]. For example, the O-O bond in PAA can be directly broken by UV radiation to generate the radicals R-O^·^ and HO^·^, thus improving disinfection efficiency and the degradation of organic compounds [52]. UV irradiation has been used to activate PAA to form active radicals that degrade naproxen (NAP). This process would be impracticable without sufficient UV intensity, because the penetration of UV light in water is limited [50]. Hu et al. investigated an advanced oxidation technique based on UV/PAA to degrade steroid estrogens Hu, Li, Zhang, et al. [53]. The metal activators of PAA include metal ions (Cu^2+^, Co^2+^, Fe^2+^, and Mn^2+^) [54,55] and metal oxides (ZVCo, Co_2_O_3_, CoFe_2_O_4_, and Co_3_O_4_) [56]. The mechanism of PAA activation by chemical activators can be triggered through the generation of organic radicals CH_3_C(O)O^·^ and CH_3_C(O)OO^·^ (Figure 2); these radicals can degrade organic pollutants by advanced oxidation. Table 2 summarises the degredation of organic pollutants by chemical activation of PAA as reported in the literature.

### 3.3. Enzymatically Generated PAA

To meet the principle of green chemistry, enzyme-generated PAA has outstanding advantages over commercial and chemically activated PAA. It is a simple, safe, low-cost, and in-situ PAA production method that avoids hazards during storage and transportation [60]. Perhydrolases are critical factors for enzyme-generated PAA, and the most commonly used ones include *Pseudomonas fluorescens esterase* [20], *acetyl xylan esterase* [61], and lipase. Perhydrolases can catalyze H_2_O_2_ and acetic acid/ethyl acetate for in-situ generation of PAA [49]. Bernhardt et al. reported that the catalytic domain of perhydrolases was Ser-His-Asp Bernhardt, Hult and Kazlauskas [62]. Table 3 summarizes the perhydrolase-producing strains used for enzyme-generated PAA in the literature. Strains-producing perhydrolases are wild microorganisms (*Pseudomonas fluorescens, Candida rugosa*, *Aspergillus niger*, *Porcine pancreas*, *Bacillus subtilis* CICC 20034, *Pichia pastoris*) and recombinant strains (*Escherichia coli* BL21, *Aspergillus ficcum*). In comparison with commercial PAA, the advantages of enzyme-generated PAA in biomass fractionation are: (1) PAA can be generated as needed, thus eliminating storage-related problems of explosion and stability. (2) Acetyl groups in biomass can be used to generate PAA. (3) PAA will sterilize the biomass to protect it from microbial contamination in biomass storage and fermentation.

## 4. Advantages of PAA Pretreatment in Lignocellulose Biorefinery

### 4.1. Helpful for Fractionation and Cellulose Saccharification

PAA pretreatment of lignocellulose can fractionate and depolymerize most of the lignin and hemicellulose, while leaving the cellulose fraction almost intact [66]. Once lignocellulose has been pretreated with PAA, high accessibility of enzyme to cellulose is achieved, and the resultant cellulose is easily hydrolyzed to release glucose. In addition, PAA pretreatment can remove most of the lignin, leading to a decrease in the effectiveness of the enzyme’s binding to lignin.

Some excellent studies on PAA pretreatment with or without catalysts in lignocellulose biorefinery are available in the literature. For instance, oil palm empty fruit bunch (OPEFB) was pretreated with 200 mM PAA in combination with 100 mM H_2_SO_4._ After pretreatment, 81.3% of the lignin was removed and 88.5% of the cellulose was retained. Experiments on enzymatic saccharification revealed that a cellulose digestion efficiency of 77.0% was achieved after PAA pretreatment, which was 1.8- and 11.9-times higher than that obtained with H_2_SO_4_ pretreatment and raw OPEFB, respectively [67]. In another paper, sugarcane bagasse was pretreated with 2% PAA and 0.1 mol/L FeCl_3_, and it was found that 57.3% of the lignin and 72.2% of the xylan were effectively removed and about 97% of the cellulose was retained. The PAA pretreated bagasse resulted in a release of 313.0 mg/g-biomass of glucose, which was 4.5 times higher than that of the untreated bagasse (69.75 mg/g-biomass) [47]. Table 4 summarizes the fractionation effectiveness of lignocellulose biomass pretreated by PAA with or without the addition of additives, as reported in the literature. Recently, a self-generated PAA oxidant in a PHP (phosphoric acid and hydrogen peroxide) pretreatment system was investigated, in which the acetyl groups in biomass played a critical role [68]. The mechanism of self-generation of PAA and the fractionation of lignocellulose in the PHP system is shown in Figure 3. The removal efficiency of lignin and hemicellulose was high—up to 83.5% and 85.7%, respectively, while 87% of cellulose was retained. Overall, PAA pretreatment with or without additives is a potentially promising proposal for the fractionation of lignocellulose biomass.

### 4.2. Beneficial to Lignin Valorization

Lignin valorization is of great importance for lignocellulose biorefinery. During PAA pretreatment, PAA acts as an advanced oxidizing agent forming free radicals, which can effectively depolymerize lignin to high value-added low molecular-mass phenolic compounds. For instance, dilute acid pretreated corn stover lignin (DACSL) and steam-exploded spruce lignin (SESPL) were treated with PAA and yielded selectively hydroxylated monomeric phenolic compounds (MPC-H) with a yield of 18% and monomeric phenolic acid compounds (MPC-A) with a yield of 22%, respectively [46]. These high value-added MPC compounds were 4-hydroxy-2-methoxycresol, p-hydroxybenzoic acid, vanillic acid, butyric acid, and 3,4-dihydroxybenzoic acid. The reaction pathway for lignin oxidative depolymerization by PAA was the Baeyer–Villiger oxidation of ketones, formed through the oxidation of benzyl hydroxyl groups adjacent to the β-O-4 linkage. Using DACSL as an example, PAA oxidation modified the side chains of hydroxyl groups, not only reducing the possibility of inter- and intramolecular hydrogen bond formation but also converting the hydroxyl groups into larger functional groups (e.g., carboxylic acids). This modification impedes π-π interaction and disrupts the integrated stacking structure of lignin (Figure 4). Therefore, the depolymerization pathway of DACSL in the presence of PAA includes side-chain replacement and side-chain oxidation (Figure 4). PAA-induced depolymerization of lignin has become a promising strategy for lignin valorization.

### 4.3. Improvement in Biomass Durability

Pathogen contamination of biomass has generally been a neglected topic in biomass biorefinery [71]. Once biomass has been contaminated by microbes during storage and fermentation, the reducing sugars are lost. Therefore, improvement in biomass durability has become an interesting topic within lignocellulose biorefinery. Chen et al. reported densifying lignocellulose biomass with alkaline chemicals (DLC) pretreatment for biomass biorefinery; they found that the densified biomass was highly resistant to microbial contamination Chen, Yuan, Chen, et al. [72]. Similarly, PAA is an organic peroxide with a wide range of antibacterial activities [73]. It can destroy the DNA and membrane lipids of microbes through the production of reactive oxygen species. PAA is effective in reducing pathogens, solid odors, and sludge [74]. It is conceivable that PAA-treated biomass will be protected from microbial contamination during storage, which will improve its durability and saccharification [75].

The relatively high cost and low safety of chemically synthesizing PAA in the laboratory limits the application of PAA pretreatment in biomass fractionation. In contrast, the development of in-situ production of PAA by bioenzymes could effectively reduce the cost. Furthermore, the disinfection and sterilization properties of PAA may be of benefit in the storage of lignocellulosic biomass.

## 5. Mass Balance and Techno-Economic Evaluation of PAA Pretreatment Technology

### 5.1. Mass Balance Analysis

Mass balance analysis is crucial for scaling up the production of PAA pretreatment technology. The procedure for converting lignocellulosic biomass to biofuels is divided into three main steps: (1) pretreatment of biomass; (2) enzymatic hydrolysis to fermentable sugars; and (3) fermentation of sugars to biofuels and subsequent distillation [76]. Mass balance covers the whole lifecycle of the biomass biorefinery process, especially the composition variances in each step throughout the pretreatment, saccharification, and fermentation [77]. Duncan et al. extensively investigated the mass balance of PAA pretreatment and saccharification of milled aspen biomass Duncan, Jing, Katona, et al. [49]. As shown in Appendix A, under PAA pretreatment with the addition of 125 mM NaOH, 1 kg of milled aspen can lead to 877 g of residual solid after 22% of the lignin and 21% of the hemicellulose are removed in the process. During saccharification, 877 g of residual solid can yield 69.6 L sugar liquid and 324 g solid. Wen et al. compared the variances of mass balance for the hydrogen peroxide-acetic acid (HPAA) and hydrogen peroxide-ethyl acetate (HPEA) pretreatment of poplar wood Wen, Chu, Zhu, et al. [70]. During the HPEA pretreatment, 677 g of holocellulose-enriched residue was obtained from 1000 g of poplar wood. In this step, 97.4% of the lignin was removed, while 90.6% of the cellulose and 81.4% of hemicellulose were recovered, respectively. After saccharification, 551.8 g of reducing sugars (including 419 g of glucose and 132.8 g of xylose) were obtained from 1000 g of raw poplar. However, HPAA pretreatment yielded only 345.2 g of reducing sugars (including 250.7 g of glucose and 94.5 g of xylose) from poplar biomass. This indicates that the HPEA pretreatment was superior to the HPAA process, as the former exhibited higher selective delignification ability and higher carbohydrate retention, as well as better digestibility. In addition, ethyl acetate is insoluble in H_2_O_2_ solution and has a lower boiling point (77 °C) than acetic acid (117.9 °C), therefore, the separation and reuse of ethyl acetate in HPEA solution is much easier than acetic acid in HPAA solvent. Yin et al. investigated a detailed mass balance for PAA pretreatment of poplar wood biomass from PAA formation, pretreatment, and saccharification Yin, Jing, Aldajani, et al. [20]. Figure 5 shows the calculated values for the mass balance of inputs, outputs, and waste. During PAA pretreatment, approximately 151 g of biomass was lost from 1 kg of poplar wood, including 57% of the lignin, 10% of the cellulose, and 13% of the hemicellulose. During enzymatic saccharification, 473 g of glucose and 148 g of xylose were released, yielding an 88.5% conversion rate from cellulose to glucose and a 73.2% conversion rate from xylan to xylose.

### 5.2. Techno-Economic Assessment

To evaluate the feasibility of PAA pretreatment’s commercialization, the technological innovation, capital, and market demand issues related to the target technology should be considered [78]. Techno-economic assessment (TEA) is an important tool for achieving the desired goal [79]. TEA consists of two main aspects: industrial design and process analysis [79], including assessing the technical feasibility, capital cost, operating cost, return on investment, payback period, and profitability [80]. Techno-economic assessment and process design are key factors for the successful and sustainable use of lignocellulose biorefinery [81]. Techno-economic assessments of PAA pretreatment in biomass biorefinery are available in the literature. For instance, the US Department of Energy’s Biomass Program conducted an economic analysis of the conversion of PAA pretreated hardwoods to bioethanol. The ethanol cost was estimated to be US$18/L for 35 wt% PAA treatment and a theoretical maximum conversion of 346 L of ethanol per dry metric ton of hardwood biomass was achieved [49]. Song et al. estimated the cost of producing bioethanol from conventional and sequential fermentation after enzymatic saccharification of hydrogen peroxide–acetic acid (HPAC) pretreated hardwoods Song, Cho, Park, et al. [82]. The cost of monosaccharides produced by HPEA pretreatment and enzyme hydrolysis was about $2.597/kg (Table 5); this was calculated from the cost of biomass (poplar), chemicals (hydrogen peroxide, acetic acid, sulfuric acid, and cellulase), and electricity (enzymatic digestion and pretreatment). Ethyl acetate is easier to separate and reuse than acetic acid in the HPAC solution. If a large number of chemicals are used in HPEA pretreatment, it increases the cost and limits the practical application of this method. Therefore, future exploration of processes with low HPEA loadings and its recycling is needed.

## 6. Challenges and Roadmap of PAA Pretreatment to Lignocellulose Fractionation

### 6.1. Immobilization of Perhydrolases for Generation of PAA

Enzyme-generated PAA has attracted much attention due to its safety and environmentally friendly green credentials. To reduce the operational cost, immobilization can be used to improve the catalytic stability and durability of the enzyme [83]. Moreover, immobilized enzymes are more conducive to the separation of enzymes from reaction substrates and products, and can be reused [84]. Recombinant acetylxylan esterase (rAXE) can be immobilized on graphite oxide (GO) to generate PAA. The immobilized rAXE shows high activity, at 62.53 U/g, and can produced approximately 134 mM of PAA. Immobilized rAXE has good stability after 10 cycles, and it maintains more than 50% of the initial yield [65]. In another study, rAXE from *Aspergillus ficcum* was immobilized on magnetic Fe_3_O_4_ chitosan nanoparticles (Fe_3_O_4_-CSN) covalent with glutaraldehyde for producing PAA [85]. In comparison with free rAXE, the immobilized rAXE exhibited better stability in the thermal and pH ranges. The immobilized rAXE showed satisfactory stability with ~90% of its activity in the aqueous phase after 10 repetitions. rAXE in *Escherichia coli* BL21 was immobilized on acrylate amino resin for PAA production; the activity of the immobilized rAXE was 383.7 U/g. It has been shown that 1 g/mL of immobilized recombinant acetyl xylan esteraser (AXE) can generate approximately 142.5 mM of PAA, and it still yields approximately 95.5 mM PAA after 10 cycles of utilization [64]. The selection of suitable carriers, improvement in activity, and the development of novel methods for immobilizomg perhydrolases represent the major challenges for enzyme-generated PAA production in the future.

### 6.2. PAA Generated In Situ Using Acetyl Groups in Lignocellulose

The formation of PAA requires acetic acid or ethyl acetate as substrate. Lignocellulosic biomass is rich in acetyl groups. Acetylation is one of the main obstacles to the effective enzymatic conversion of hemicellulose to fermentable sugars. Using these acyl groups to produce PAA in situ is not only beneficial to the hydrolysis of hemicellulose but also helpful in reducing the cost of PAA. Tian et al. investigated self-generation PAA in a phosphoric acid plus hydrogen peroxide system Tian, Chen, Shen, et al. [68], describing the overall deconstruction of lignocellulose and degradation of hemicellulose/lignin. Further experiments on the basic and practical application of self-generation PAA for lignocellulose biorefinery should be conducted in the future.

### 6.3. Synergistic Effect of Additives and PAA

PAA pretreatment offers effective delignification during lignocellulose fractionation [70]. To increase the digestibility of biomass, PAA pretreatment of lignocellulose has been performed and the combined hydrothermal, sonication, catalysts, acids, bases, ionic liquids, and other chemical reagents evaluated. Pretreatment of biomass using heat-assisted PAA at 90 °C for 5 h achieved 90% delignification and increased the digestibility of treated hardwood and softwood biomass by 32% and 23%, respectively [86]. When the biomass was treated with hot compressed water and enzyme-generated PAA, 90% of hemicellulose and 70% of lignin were removed. The cellulose residue released 90% glucose [63]. Orange bagasse was treated with ultrasound at 30% amplitude for 10 min followed by PAA treatment for 24 h; 81.49% of the cellulose was retained and almost the hemicellulose (99.12%) and lignin (97.32%) were removed [87]. Lewis acid can destroy lignocellulose structure and increase the accessibility of PAA to lignin. When sugarcane bagasse was treated by PAA and FeCl_3_, hemicellulose depolymerized into monosaccharides without cellulose destruction [47]. When corn stover was treated with 1.5 wt% PAA and 3 wt% maleic acid at 130 °C for 1 h, 86.83% of the cellulose was retained and 88.21% of the hemicellulose and 87.77% of the lignin were dissolved in the aqueous liquid (Figure 6). Enzymatic digestion of the cellulose-rich fraction has been shown to release 89.65% of glucose, which is more than two times higher than with the untreated substrate [66]. Delignification efficiency can be greatly increased by the combination of PAA and alkali treatment [88]. Alkali-assisted PAA pretreatment has been employed to treat sugarcane bagasse for enzymatic digestion, for the production of ethanol by simultaneous saccharification fermentation (SSF), and for the further conversion of xylose to 2,3-butanediol. Results showed that approximately 45 g/L ethanol (0.30 g ethanol/g pulp, 68.6% theoretical yield) and 0.35–0.50 g 2,3-butanediol were obtained [89]. PAA combined with ionic liquid pretreatment has been applied to pine wood to enhance enzymatic saccharification of cellulose by 45–70% [90]. In future, it is expected that a greener, more efficient, and lower-cost PAA pretreatment system will be developed for lignocellulose fractionation.

## 7. Conclusions and Prospects

In summary, PAA pretreatment has proven an ideal and promising strategy for lignocellulose biorefinery. In this article, three methods of PAA pretreatment were reviewed, each of them with its own merits and shortcomings. From the perspective of green chemistry, enzyme-generated PAA for lignocellulose fractionation should attract the most attention. To evaluate the feasibility of the PAA pretreatment process, the mass balance and techno-economic analysis of PAA pretreatment were investigated. Although many breakthroughs have been achieved in PAA pretreatment for lignocellulose biorefinery, some prospective developments can be proposed for the future:(1)The use of acetyl groups in lignocellulose to replace chemical ethyl acetates should be developed for the self generation of PAA.(2)The use of perhydrolase-producing microbes should be broadened, and the activity and selectivity of perhydrolases enhanced. Furthermore, novel techniques for the immobilization of perhydrolases should be investigated to increase enzyme solvent durability.(3)A multi-functional system in combination with PAA and other chemical or physical intensification should be established. Through the integrated PAA pretreatment system, the stubborn structure of biomass can be easily disrupted to achieve high delignification.

## Figures and Tables

**Figure 1 molecules-27-06359-f001:**
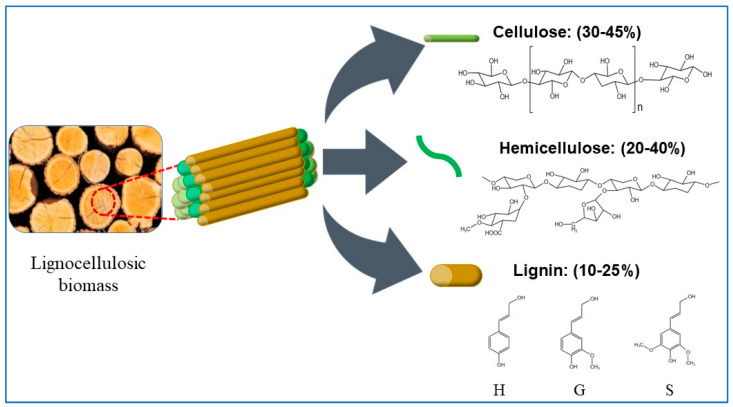
Structural compositions of lignocellulosic biomass.

**Figure 2 molecules-27-06359-f002:**
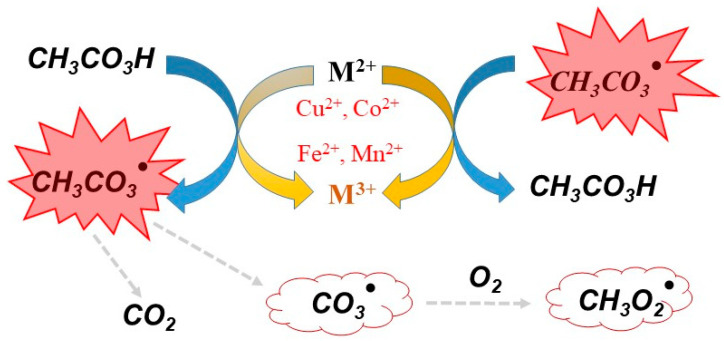
Free radicals generated by PAA in the presence of the metal ions activators (Cu^2+^, Co^2+^, Fe^2+^, and Mn^2+^).

**Figure 3 molecules-27-06359-f003:**
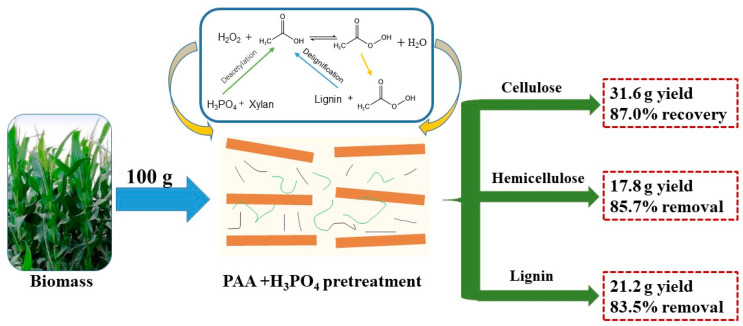
Self-generation of PAA in PHP (phosphoric acid plus hydrogen peroxide) system mediating overall deconstruction of lignocellulose and degradation of hemicellulose/lignin according to the data from reference [69].

**Figure 4 molecules-27-06359-f004:**
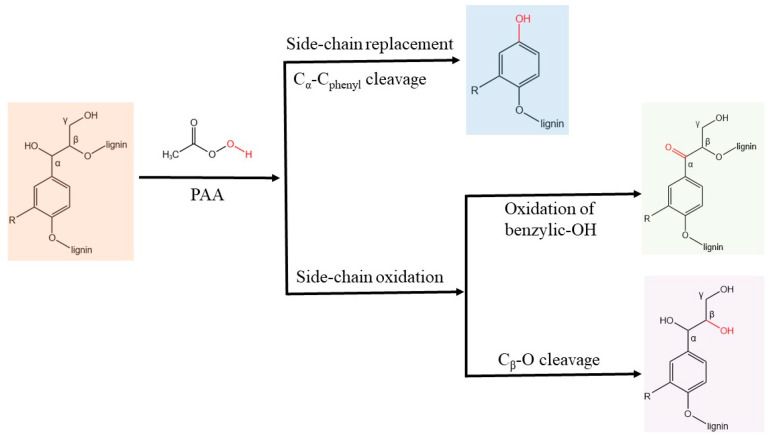
The depolymerization pathway of dilute acid pretreated corn stover lignin (DACSL) by PAA.

**Figure 5 molecules-27-06359-f005:**
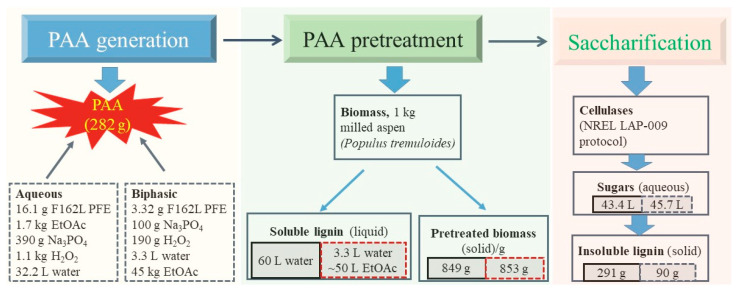
Mass balance with PAA pretreatment of milled poplar wood, including PAA formation, pretreatment and saccharification processes according to the data from reference [20].

**Figure 6 molecules-27-06359-f006:**
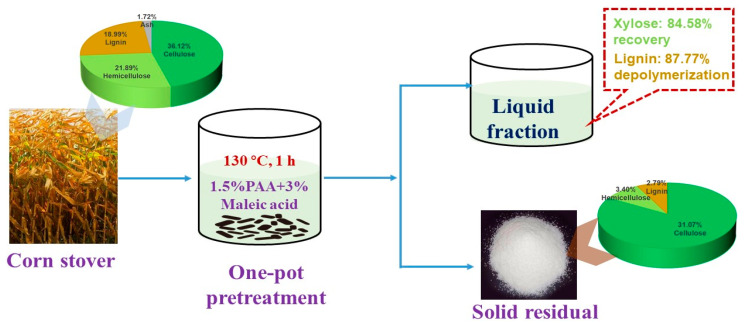
Lignocellulose fractionation of corn stover treated by 1.5 wt% PAA combined with 3 wt% maleic acid according to the data from reference [66].

**Table 1 molecules-27-06359-t001:** Detailed information of part commercial PAA products in the literature [48,49].

Identity	Product Name	Supplier and Country	PAA(%)	H_2_O_2_(%)	PAA:H_2_O_2_
Lspez	Wofasteril L. Spez	KESLA PHARMA WOLFEN GmbH (Greppin, Germany)	3	40	0.034
E35	Wofasteril 035	KESLA PHARMA WOLFEN GmbH (Greppin, Germany)	3.5	10	0.156
SC50	Wofasteril SC50	KESLA PHARMA WOLFEN GmbH (Greppin, Germany)	5	8	0.28
AC150	Peressigsaure 15% reinst	Applichem GmbHt (Darmstadt, Germany)	15	24	0.28
E250	Wofasteril E250	KESLA PHARMA WOLFEN GmbH (Greppin, Germany)	25	30	0.37
S1400	Sigma-Aldrich Peracetic Acid Solution	Sigma-Aldrich Co. (St. Louis, MO, USA)	39	6	2.91
E400	Wofasteril E400	KESLA PHARMA WOLFEN GmbH (Greppin, Germany)	40	12	1.49
S1400	Sigma-Aldrich 32 wt% PAA	Sigma-Aldrich Co. (St. Louis, MO, USA)	32	5	6.4
/	/	Thermo Fisher Scientific (New York, NY, USA)	39	/	/
VigorOx^®^ WWTII	PAA technical grade solution (VigorOx^®^ WWTII)	PeroxyChem (Philadelphia, Pennsylvania, USA)	15	23	0.652

**Table 2 molecules-27-06359-t002:** Degradation of organic pollutants by chemical activation PAA.

Compounds	Chemical Activator (Catalyst)	Compounds Concentration	Conditions: Temperature, pH, Catalyst Loading	PAA Concentration	Degradation Rate (%)	References
Orange G	Co_3_O_4_	0.05 mM	25 °C, 7.0, 0.1 g/L	0.5 mM	100	[57]
Sulfamethoxazole	CoFe_2_O_4_	10 μM	23 °C, 7.0, 0.1 g/L	100 μM	87.3	[56]
Bisphenol-A	Co (II)/Co (III)	15 μM	22 °C, 4.0, 10 μM,	100 μM	100	[58]
Carbamazepine					87.7	
Naproxen					100	
Sulfamethoxazole					98.5	
Sulfamethoxazole	Co	10 μM	25 °C, 7.0, 0.8 μM	100 μM	89.4	[51]
Naproxen	UV	4 μM	20 °C, 7.0, /no catalyst	20 mg/L	100	[50]
Bisphenol-A	Fe (II)	15 μM	22 °C, 6.0, 5 μM,	20 μM	87.7	[55]
Methylene blue					89.4	
Naproxen					98.2	
Sulfamethoxazole	ZVCo *	5 μM	25 °C, 7, 0.1 g L^−1^	50 μM	99.4	[59]
Steroid estrogens	UV	50 μg/L	25 °C, 6.01, /no catalyst	30 mg/L	90	[53]

* ZVCo: zero-valent cobalt.

**Table 3 molecules-27-06359-t003:** Perhydrolases producing strains and enzyme-generated PAA.

Perhydrolase	Strains	Reagent Dosage (EA/GT, H_2_O_2_)	Conditions: Temperature, pH, Enzyme Loading	PAA Concentration (mM)	References
*Pseudomonas fluorescens* esterase (PFE)	*Pseudomonas fluorescens*	500 mM EA *, 1.0 M H_2_0_2_	23 °C, 7.2, 0.5 mg/mL	115	[20]
*Pseudomonas fluorescens* esterase (PFE)	*Escherichia coli* BL21	500 mM EA *, 1.0 M H_2_0_2_	23 °C, 7.2, 0.5 mg/mL	90	[63]
PFE-L29G	*Pseudomonas fluorescens*	600 mM EA *, 500 mM H_2_0_2_	37 °C, 7.0, 0.5 mg/mL	60	[49]
Wild-type PFE	*Pseudomonas fluorescens*			70	
Lipase Type VII	*Candida rugosa*	250 mM GT †, 1.0 M H_2_0_2_	25 °C, 7.4, 0.6 mg/mL	0.98	[64]
LPL	*Aspergillus niger*			2.6	
Lipase Type II	*Porcine pancreas*			7.1	
Acetylxylan esterase (AXE)	*Bacillus subtilis CICC 20034*			113.37	
Acetylxylan esterase (AXE)	*Pichia pastoris*	0.5 M EA *, 1.0 M H_2_0_2_	37 °C, 7.0, 15 mg/mL	133.70	[65]
Recombinant acetylxylan esterase (rAXE)	*Aspergillus ficcum*	500 mM EA *, 1.0 M H_2_0_2_	37 °C, 7.0, 0.1 mg/mL	134	[61]

* EA: Ethyl acetate; † GT: Glycerol triacetate.

**Table 4 molecules-27-06359-t004:** PAA pretreatment for lignocelluloses fractionation and its effectiveness.

Lignocellulose Biomass	PAA Treatment Conditions	Cellulase Loading	Lignin Removal Rate (%)	Hemicellulose Removal Rate (%)	Cellulose Retaining Rate (%)	Saccharification of Cellulose (%)	References
Oil palm empty fruit bunch	Solid loading 1:20, 200 mM PAA, 100 mM H_2_SO_4_, 140 °C, 5 min	30 U/g	81.3	88.5	81.1	77	[67]
Sugarcane bagasse	Solid loading 1:10, 2 wt% PAA, 90 °C, 60 min, 250 rpm	20 FPU/g	40.6	58.2	93.4	48.78	[47]
Wheat straw	Solid loading 1:10, 65 g H_3_PO_4_, 7.1 g H_2_O_2_, 50 °C, 5 h, 180 rpm	/	90	100	87	/	[68]
Yellow poplar	Solid loading 1:50 300 mM PAA, 100 mM H_2_SO_4_, 120 °C, 5 min, 180 rpm	30 FPU/g	90.4	85.7	75.6	84.0	[69]
Poplar	Solid loading 1:10, 1:1 (*v*/*v*) H_2_O_2_ (30%): GAA *, 80 °C, 2 h,	5 FPU/g	94.1	26.6	98.7	52.7	[70]
	Solid loading 1:10, 1:1 (*v*/*v*) H_2_O_2_ (30%): EA † (99%), 80 °C, 2 h,		97.2	17.0	95	90.6	
Corn stover	Solid loading = 1:40, 1.5 wt% PAA, 3 wt% MA ‡, 130 °C, 1 h	10 FPU/g	87.77	88.21	86.83	89.65	[50]

* GAA: Glacial acetate acid; † EA: Ethyl acetate; ‡ MA: Maleic acid.

**Table 5 molecules-27-06359-t005:** HPEA pretreatment poplar monosaccharide production cost estimation.

	Material and Process	Cost, $/kg Monosaccharides
Biomass	Poplar	0.057
Chemical	Hydrogen peroxide	0.512
	Ethyl acetate	1.211
	Acetic acid	0.906
	Sulfuric acid	0.001
	Cellulase	0.284
Electricity	Enzymatic hydrolysis	0.532
	HPEA pretreatment	0
	HPAA pretreatment	0

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
