# Peer review of "Peroxyacetic Acid Pretreatment: A Potentially Promising Strategy towards Lignocellulose Biorefinery"

_molecules, 2022, doi:10.3390/molecules27196359_

Round 1
Reviewer 1 Report
1. Line 57 - it is recommended to write the chemical formulas of carbon dioxide and sulfur dioxide.
2. Line 73. Change the words to: Therefore, more researchers are ....
3. Line 125. Delete the second 'the'
4. Table 1. There must be a reference for this Table.
5. Table 2. What is ZVCo? Is it Zero valent cobalt?
6. Line 165. The 'P' for pseudomonas should be capitalized.
7. Table 3. Spelling error for 'flfluorescencs' - Ref 18.
8. Line 236. What is DLC? Always provide a full name before using an abbreviation.
9. Line 309. What is rAXE? Write its full name first.
Author Response
1) Line 57 - it is recommended to write the chemical formulas of carbon dioxide and sulfur dioxide.
Response: Thank you for your valuable comments. We have revised the writing and corrected it in the revised version. The chemical formulas of carbon dioxide and sulfur dioxide has been corrected to CO2 and SO2, respectively.
2) Line 73. Change the words to: Therefore, more researchers are ....
Response: Thank you for your good comments. We have changed the words to “Therefore, more researchers are….” in the revised version.
3) Line 125. Delete the second 'the'
Response: Thank you for your valuable comments. We have Deleted the second 'the'.
4) Table 1. There must be a reference for this Table.
Response: Thank you for your valuable comments. We have cited the references in Table 1 in revised version.
5) Table 2. What is ZVCo? Is it Zero valent cobalt?
Response: Thank you for your valuable comments. ZVCo is Zero valent cobalt, and we have added a note for ZVCo in the end of Table 2.
6) Line 165. The 'P' for pseudomonas should be capitalized.
Response: Thank you very much for your valuable comment. We have corrected it as follows. Perhydrolases are the critical factors for enzyme-generated PAA, which mainly include Pseudomonas fluorescens esterase, acetyl xylan esterase and lipase.
7) Table 3. Spelling error for 'flfluorescencs' - Ref 18.
Response: Thank you for your valuable comments. We have corrected 'flfluorescencs' in Table 3 to 'fluorescencs'.
8) Line 236. What is DLC? Always provide a full name before using an abbreviation.
Response: Thank you for your valuable comments. DLC is the abbreviation of “densifying lignocellulose biomass”. We have provided the full name of DLC in the revised version.
9) Line 309. What is rAXE? Write its full name first.
Response: Thank you very much for your valuable guidance. We have added the full name of rAXE. Which is the abbreviation of “recombinant acetyl xylan esterase”.
Reviewer 2 Report
The manuscript entitled “Peroxyacetic acid pretreatment: A potential promising strategy towards lignocellulose biorefinery” by Hu et al., reports the impact of peroxyacetic acid pretreatment on the downstreaming process of lignocellulose into biofuels. The manuscript presents an interesting and important topic related to biorefinery. Although, authors have attempted a nice study; however, it remains with some problems particularly the organization and language of the manuscript. Following are some of the major issues that need to be corrected for possible publication. Therefore, I suggest the authors to revise the manuscript after careful considerations.
Abstract
1. Author can write a statement for objectives as well as conclusion of the manuscript.
2. What is meant by “exits”?
Introduction
1. Line 43, Correct the statement, “cellulose to cellulose”.
2. Line 60, Start a new sentence from, “it is a novel..”.
3. The link between paragraphs and the flow of information is missing in the manuscript. Many of the paragraphs start all of a sudden without links with the previous paragraphs.
4. The literature referred to for the current analysis is meager and needs to be elaborated. Authors have a scope to go through the following recent references and cite them in the manuscript, besides others.
A. Valorization potential of a novel bacterial strain, Bacillus altitudinis RSP75, towards Lignocellulose Bioconversion: An assessment of symbiotic bacteria from the stored grain pest, Tribolium castaneum. Microorganisms 9, 1952. https://doi.org/10.3390/microorganisms9091952
B. Evaluation and characterization of the cellulolytic bacterium, Bacillus pumilus SL8 isolated from the gut of oriental leafworm, Spodoptera litura: an assessment of its potential value for lignocellulose bioconversion. Environmental Technology & Innovation 27 (2022) 102459. https://doi.org/10.1016/j.eti.2022.102459
C. Purification of a cellulase from cellulolytic gut bacterium, Bacillus tequilensis G9 and its evaluation for valorization of agro-wastes into added value byproducts. Biocatalysis and Agricultural Biotechnology 20 (2019)101219. https://doi.org/10.1016/j.bcab.2019.101219
5. One of the major flaws of the present study is the lacunae of the critical analysis of the available literature. The authors have just compiled a summary of different studies that are already available in the literature. The beauty of the review articles is to highlight the drawbacks of the current methods and research and propose a roadmap for future studies that can be undertaken to address a major goal. Therefore, the authors have a scope to improve the article by shedding some light on the critical evaluation of each aspect presented in the manuscript.
Author Response
Abstract
1) Author can write a statement for objectives as well as conclusion of the manuscript.
Response: Thank you for your valuable comments. The abstract has been re-organized, and it includes objectives as well as conclusion of the manuscript in the revised version.
2) What is meant by “exits”?
Response: Thank you for your valuable comments. We have rewritten the sentence and deleted the word "exits" in the revised version.
Introduction
1) Line 43, Correct the statement, “cellulose to cellulose”.
Response: Thank you for your valuable comments. We have corrected “…the accessibility of cellulose to cellulose” to “… the accessibility of cellulase to cellulose”.
2) Line 60, Start a new sentence from, “it is a novel..”.
Response: Thank you for your valuable comments. We have started a new sentence from, “It is a novel….” In the revised version.
3) The link between paragraphs and the flow of information is missing in the manuscript. Many of the paragraphs start all of a sudden without links with the previous paragraphs.
Response: Thank you for your valuable comments. We have carefully checked the whole the manuscript and added paragraph-to-paragraph linking words in the revised version, so that the revised manuscript is more readable than original form.
4) The literature referred to for the current analysis is meager and needs to be elaborated. Authors have a scope to go through the following recent references and cite them in the manuscript, besides others.
- Valorization potential of a novel bacterial strain, Bacillus altitudinis RSP75, towards Lignocellulose Bioconversion: An assessment of symbiotic bacteria from the stored grain pest, Tribolium castaneum. Microorganisms 9, 1952. https://doi.org/10.3390/microorganisms9091952
- Evaluation and characterization of the cellulolytic bacterium, Bacillus pumilus SL8 isolated from the gut of oriental leafworm, Spodoptera litura: an assessment of its potential value for lignocellulose bioconversion. Environmental Technology & Innovation 27 (2022) 102459. https://doi.org/10.1016/j.eti.2022.102459
- Purification of a cellulase from cellulolytic gut bacterium, Bacillus tequilensis G9 and its evaluation for valorization of agro-wastes into added value byproducts. Biocatalysis and Agricultural Biotechnology 20 (2019)101219. https://doi.org/10.1016/j.bcab.2019.101219
Response: Thank you for your valuable comments. The three articles mentioned above have been cited in the revised version.
5) One of the major flaws of the present study is the lacunae of the critical analysis of the available literature. The authors have just compiled a summary of different studies that are already available in the literature. The beauty of the review articles is to highlight the drawbacks of the current methods and research and propose a roadmap for future studies that can be undertaken to address a major goal. Therefore, the authors have a scope to improve the article by shedding some light on the critical evaluation of each aspect presented in the manuscript.
Response: Thank you very much for your good comments. The drawbacks of the current methods and research have been provided in the revised version. Simultaneously, some critical evaluation of each aspect presented in the manuscript has also highlighted in the revised version.
Round 2
Reviewer 2 Report
The authors have dressed all the comments and made substantial changes to the manuscript, I recommend it for possible publication in its current form by the journal.